# Phytochemicals, Antioxidant Attributes and Larvicidal Activity of *Mercurialis annua* L. (*Euphorbiaceae*) Leaf Extracts against *Tribolium confusum* (Du Val) Larvae (*Coleoptera*; *Tenebrionidae*)

**DOI:** 10.3390/biology10040344

**Published:** 2021-04-20

**Authors:** Rania Ben Nasr, Elie Djantou Baudelaire, Amadou Dicko, Hela El Ferchichi Ouarda

**Affiliations:** 1Laboratoire de Chimie et de Physique—Approches Multi-échelles des Milieux Complexes (LCP-A2MC)1, Boulevard Arago, CEDEX 03, 57078 Metz, France; amadou.dicko@univ-lorraine.fr; 2Laboratory of Plant Toxicology & Environmental Microbiology, Faculty of Science of Bizerte, University of Carthage, Zarzouna 7021, Tunisia; helaelferchichi.ouarda@gmail.com; 3AGRITECH, 4, Rue Piroux, 54000 Nancy, France; elie.baudelaire@agritech-france.fr

**Keywords:** *Mercurialis annua* L., antioxidant activity, LC-MS, larvicidal activity

## Abstract

**Simple Summary:**

*Mercurialis annua* L. is a *Euphorbiacea* widespread in the Euro-Mediterranean region. This species is known for its toxicity and is used in traditional pharmacopoeia. However, the chemical composition of the aerial parts of this taxon and their larvicidal impact remain to be explored. However, the literature reports only a few studies on the relationship between the polyphenolic profile of the different extracts of *Mercurialis annua* L. and the larvicidal activity; therefore, we tested different extracts of *M*. *annua* against the larvae of *Tribolium confusum* (Du Val) for better protection of stored foodstuffs attacked by this pest. On the other hand, green chemistry as a distinct discipline has developed in recent years, which allowed us to identify different active compounds by LC-MS, and study the relationship between larvicidal activity and the chemical composition of extracts from male and female leaves of *Mercurialis annua* L.

**Abstract:**

This study reports the link between larvicidal activity and the phytochemical composition of male and female leaf extracts of *Mercurialis annua* L. (*M*. *annua*) from four Tunisian regions: Bizerte, Jandouba, Nabeul and Beja. Their antioxidant activity was evaluated using DPPH (2, 2-diphenyl-1-picrylhydrazyl) assays. Phenolic compounds were identified and quantified using liquid chromatography coupled with a UV detector and mass spectrometry (LC-UV-ESI/MS). Higher antioxidant activity (AOA) was found in the leaves of male plant extracts than of female ones. The leaves of male and female plant extracts from Bizerte exhibited the highest AOA: 22.04 and 22.78 mg Trolox equivalent/g dry matter (mg TE/g DM), respectively. For both sexes, plant extracts from Beja had the lowest AOA with 19.71 and 19.67 mg TE/g DM for male and female plants, respectively. Some phenolic compounds such as narcissin, gallocatechin, rutin, epigallocatechin and epicatechin were identified and quantified using LC-MS, which highlighted the abundance of narcissin and rutin in the male leaves of *M*. *annua*. We noted that the interaction between the sex of plants and the provenance had a significant effect on TFC (F = 6.63; *p* = 0.004) and AOA (F = 6.53; *p* = 0.004) assays, but there was no interaction between sex and origins for TPC (F = 1.76; *p* = 0.19). The larvicidal activity of aqueous leaf extracts of *M*. *annua* against *Tribolium confusum* (Du Val) (*T*. *confusum*), an insect pest of flour and cereal seeds, showed that the mortality could reach 100% after 48 h of exposure in the Bizerte region. The LC_50_ values for the leaf extract were low in Bizerte, with 0.003 and 0.009 g/mL for male and female plants, respectively, succeeded by Jandouba, which displayed 0.006 and 0.024 g/mL for male and female plants, respectively. Nabeul showed 0.025 g/mL for male plants and 0.046 g/mL for female plants and Beja showed 0.037 and 0.072 g/mL for male and female plants, respectively. This is the first time that a study has revealed a negative correlation between the antioxidant activity and the larvicidal activity of the leaf extracts of *M*. *annua* with the following correlation coefficients of Perason: r = −0.975 and r = −0.760 for male and female plants, respectively.

## 1. Introduction

*Mercurialis annua* L. (*M*. *annua*) is an annual herbaceous and dioecious species of the family *Euphorbiaceae*. It has been reported that it is a mesophilic and wind-pollinated plant, especially found in disturbed and roadside habitats throughout Central and Western Europe and in North Africa, as well as around the Mediterranean Basin [1]. The so-called Mercuriale of gardens was widely used in traditional medicine to control many diseases [2]. *M*. *annua* has laxative properties; it was used to treat pathologies of the digestive system while increasing intestinal transit and facilitating the evacuation of salts, with purgative activities. In addition, the plant was used to treat obstinate constipation and circumscribed inflammation in the intestines and stomach. It is useful against water retention and bladder infections [3]. It has diuretic properties, which increase urine production. *M. annua* is an emetic plant that causes vomiting, and it is useful to reduce gynecological diseases, such as menstrual symptoms like fatigue and headaches occurring during the menstrual cycle [4]. Research carried out in Morocco indicated that the aqueous extracts of the aerial parts of *M*. *annua* have an anxiolytic action [5]. The plant also possesses noticeable anti-microbial activities [6]. Recent studies indicated that methanolic extracts of *M*. *annua* were found to have a cytotoxic activity against cancer cells, stimulating the immune system cells to release cytokines in vitro as well as in vivo. These results clearly show that leaf extracts of *M*. *annua* display a cytotoxic effect on two cell lines, namely colorectal (HRT-18) and breast cancer (T47D) [7]. Therewith, Ostrozhenkova et al. (2007) [8] were able to isolate in *M. annua* the pyridinone-type chromogen, hermidine, and NMR spectroscopy also allowed them to isolate alanine. Biological activities research revealed an insecticide and larvicide robust potential in *Mercurialis perennis* L. and subsequently in *M*. *annua* [4]. Blanco-Salas et al. (2019) [3] stated that the genus *Mercurialis* possesses antidiabetic and antihypertensive attributes due to the affluence of its aerial parts in flavonoids rutin and narcissin. Furthermore, this species was used in traditional pharmacopoeia against hair loss in Tunisia. However, studies on the polyphenolic profile and larvicidal activity in relation to antioxidant potential of *M*. *annua* leaf extracts from male and female plants are scarce. Moreover, in the accessible literature, there is no report about the Tunisian origins and the natural extracts of *M*. *annua* to limit *Tribolium confusum* (Du Val) (*T*. *confusum*) amplification. On the other hand, our stocks of foodstuffs and especially cereals which occupy a strategic place in the diet of Tunisians have always been sensitive to attacks by *T*. *confusum*, causing a quantitative and qualitative loss of the stored food. The utilization of chemicals against this pest are a conventional method of control, but its disadvantages are numerous, citing as an example the sinister consequences for human health due to the accumulation of these chemicals in our food. According to Ghnimi et al. (2014) [9], several species of the *Euphorbiaceae* family have a detrimental effect against insect pests. For all these reasons, we planned to test different extracts of *M*. *annua* against *T*. *confusum* larvae and assess their antioxidant power, with a view to better protect stored food.

## 2. Material and Methods

### 2.1. Plant Material

*M. annua* was identified by Pottier-Alapetite (1979) [10] as a dioecious species and our samples were compared to the authentic specimen from the herbarium of the Faculty of Science of Bizerte. In fact, leaves of male and female plants of *M*. *annua* were randomly collected between December and February from four different regions of Tunisia, belonging to different bioclimatic stages, from sub-humid to semi-arid. These provenances were Bizerte (37°16′27′′ N; 9°52′26′′ E; 33 m), Béja (36°43′31′′ N; 9°11′31′′ E; 212 m), Jandouba (36°30′02′′ N; 8°46′51′′ E; 141 m) and Nabeul (36°27′15′′ N; 10°44′05′′ E; 12 m). Then, leaves were dried at room temperature (27 °C ± 2 °C) and were reduced to powder with an electrical blender and conserved at −4 °C until use.

### 2.2. Insect Larvae

The larvicidal activity was carried out on 5 mm-long and pale-yellow larvae of *T*. *confusum*, which coincides with the pre-pupal stage [11]. The animal material was maintained at ambient rearing conditions in the entomology laboratory at the National Institute of Agronomic Research of Tunisia. All tests were conducted at room temperature (27 ± 2 °C).

### 2.3. Chemicals Used

Folin–Ciocalteu reagent was obtained from VWR International (Metz, France). Gallic acid, rutin, catechin, gallocatechin, epigallocatechin, chlorogenic acid, vanillic acid, caffeic acid, epicatechin, narcissin, verbacoside, naringin, Trolox and sodium nitrite (NaNO_2_), were obtained from Sigma-Aldrich (Paris, France). Sodium carbonate (Na_2_CO_3_), aluminum chloride (AlCl_3_), and sodium hydroxide (NaOH) from Acros Organics (Brussels, Belgium).

### 2.4. Preparation of Extracts

The hydro-methanolic extracts of male and female leaves powder of *M*. *annua* were processed according to the experimental method used by Zaiter et al. (2017) [12] with some modifications. A total of 2 g of each powder was macerated in 20 mL of 70% methanol and 30% water for 24 h at room temperature with stirring. Then, extracts were centrifuged at 4000 rpm for 30 min. The supernatant was collected, filtered through 45 μm polytetrafluoroethylene (PTFE) filters and stored in a sealed dark bottle at −4 °C until analysis.

#### 2.4.1. Quantification of Total Phenolic Contents

Total phenolic contents (TPC) were determined following the method by Mousavi et al. (2019) [13], which was slightly modified. Briefly, 500 μL of Folin–Ciocalteu reagent (1 N) was added to 100 μL of each extract, which was diluted with 5 mL distilled water. After 5 min, 2 mL of Na_2_CO_3_ (7%) was added and the mixture was incubated at room temperature. After 60 min, the absorbance was measured at 725 nm against the blank (methanol). A calibration curve was prepared using different methanolic solutions of gallic acid as a standard (0.75, 1.5, 2.56, 3.01, 3.77 and 4.5 mg/mL). The results were expressed in terms of milligrams of gallic acid equivalents per gram of dry matter (mg GAE/g DM).

#### 2.4.2. Quantification of Total Flavonoids Contents

Total flavonoid compounds (TFC) were dosed by colorimetric assays described by Chen et al. (2011) [14], with slight modifications. Briefly, 100 μL of extract was added to a flask containing 6 mL of distilled water. Then, 300 μL of 5% NaNO_2_ was added to the flask. After 5 min, 300 μL of 10% AlCl_3_ was added. Finally, after 6 min, 2 mL of 1 M NaOH was added. The solution was well shaken, and the absorbance was measured at 510 nm with a UV-Visible spectrophotometer Cary 50 Scan (Agilent, Santa Clara, CA, USA). A calibration curve was prepared using different methanolic solutions of catechin as a standard (0.46, 0.63, 0.83, 1.3, 2.05 and 2.24 mg/mL). The results were expressed in terms of milligrams of catechin equivalents per gram of dry matter (mg CE/g DM).

#### 2.4.3. Evaluation of Antioxidant Activity

To evaluate the Antioxidant Activity (AOA) of *M*. *annua*, the DPPH (2, 2-diphenyl-1-picrylhydrazyl) test was used according to the protocol described by Velazquez et al. (2003) [15] and Molyneux et al. (2004) [16], with some modifications. Briefly, into tubes were introduced 100 μL of each extract and 3 mL of the methanolic solution of DPPH (2.4 mg/100 mL). After shaking, the tubes were placed in the dark at room temperature for 30 min. Reading was performed by measuring the absorbance at 517 nm. The blank was a DPPH methanolic solution. A calibration curve was prepared using different methanolic solutions of Trolox as a standard (0.1, 0.3, 0.7, 1, 1.3, 1.7 and 2 mg/mL). The results were expressed in terms of milligrams of Trolox equivalents per gram of dry matter (mg TE/g DM) [13].

### 2.5. LC-MS Analysis

LC-MS analyses were performed on a LC-MS 2020 system (Shimadzu, Tokyo, Japan) associated with an electrospray ionization source (ESI). Separation was carried out on a 150 × 4.6 m^2^ i.d. C18 reverse phase Gemini column (Phenomenex, Torrance, CA, USA) with a particle size of 3 μm and pore size of 130 Å. The column oven was fixed at 30 °C. The mobile phase consisted of (A) 0.5% formic acid in water and (B) acetonitrile. The injection volume was 20 μL and the flow rate was 0.6 mL/n min. The 40 min gradient used was as follows: 0–5 min, 90:10 (A:B); 5–7 min from 90:10 to 86:14 (linear gradient); 7–17 min held at 86:14; 17–19 min from 86:14 to 75:25 (linear gradient); 19–24 min held at 75:25; 24–25 min from 75:25 to 10:90 (linear gradient); 25–31 min held at 10:90; 31–32 min from 10:90 to 90:10; and 32–40 min held at 90:10 (re-equilibration step). The electrospray ionization source was operated in negative mode. The nebulization gas flow was set to 1.5 L/min and drying gas flow at 20 L/min. The heat block temperature was fixed at 350 °C, and the desolvation line (DL) temperature at 250 °C. The probe voltage was set to −4500 V. Phenolic standards used in this study were gallic acid, rutin, catechin, gallocatechin, epigallocatechin, chlorogenic acid, vanillic acid, caffeic acid, epicatechin, narcissin, verbacoside and naringin.

### 2.6. Evaluation of Larvicidal Activity

#### 2.6.1. Toxicity Tests

Extraction was carried out according to Aouinty et al. (2006) [17] and Soha et al. (2019) [18], with slight modifications. Indeed, the aqueous extracts were obtained from the leaves of the male and female plants of *M*. *annua*, and 10 g of each powder was macerated in 100 mL of distilled water for 24 h, then filtered using a Whatman filter paper (3 mm). The resulting filtrate represents an initial stock solution of 10%. To give more signification to the quantities of the plant material, they were concentrated by evaporation under reduced pressure using a rotary evaporator. The residues obtained were used in the larvicidal activity tests with different concentrations (0.01, 0.005, 0.002 and 0.001 g/mL). The toxicity was evaluated by placing 15 *T. confusum* larvae in a glass beaker containing 20 mL of each concentration of each extract. The control group contained 20 mL of distilled water and 15 larvae. Mortality was registered after 6, 12, 24 and 48 h of exposure and the mortality percentage was reported from an average of three replicates.

#### 2.6.2. Statistical Analysis

The lethal concentration (LC_50_) was defined as the concentration that caused 50% mortality in the population of larvae studied, for a given time [19]. The estimates of LC_50_ were obtained after 24 h and after 48 h, using probit analysis. Statistical analysis was completed using the one-way ANOVA test to compare the effects of the different parameters (concentration, time, provenance and sex) studied. Two-way ANOVA was used to estimate the interaction significance of the effects of sex and provenance. The Bonferroni test was used to investigate the differences between male and female plant assays and to compare *M*. *annua* provenances. All tests were carried out in triplicate and the results are expressed as means ± standard deviations. The correlation between TPC, TFC and antioxidant activity is presented by Pearson correlation coefficient. Results were considered statistically significant when *p* < 0.05. Statistical processes were achieved using SPSS version 25.0 software.

## 3. Results and Discussion

### 3.1. Quantification of TPC and TFC

The results reported in Figure 1 and Figure 2 show that the distribution of TPC and TFC varied according to the region (*p* < 0.05) and sex (*p* < 0.005). The TPC in leaf extracts from Bizerte was the highest, with more than 100.41 and 89.7 mg GAE/g DM for male and female plant extracts, respectively, while leaf extracts from Beja exhibited the lowest values: 44 mg GAE/g DM for female plants and 47.49 mg GAE/g DM for male plants (Figure 1).

The TFC in leaf extracts from Bizerte showed the highest concentration for both sexes: 22.5 CE mg/g DM in female plants and 39.94 CE mg/g DM in male plants. This was followed by Jandouba, with 36.94 CE mg/g DM in male plants, and Nabeul, also with 30.35 CE mg/g DM in male plants, while the provenance of Beja displayed for both male (4.27 CE mg/g DM) and female (4.16 CE mg/g DM) leaves the lowest doses (Figure 2). Richness in TFC in aerial parts of plants was previously reported by Caridade et al. (2018) [20]. This affluence in phenolic compounds is probably the cause of the use of *M*. *annua* against hair loss in Tunisia. Blanco-Salas et al. (2019) [3] clarified the effectiveness of several species of the genus *Mercurialis* as a refuge in traditional Spanish medicine.

The TPC and TFC of the leaf extracts were higher in the Bizerte region for both sexes, and they were the highest in the male plant for the provenances tested. The ANOVA test showed that there was a highly significant difference between different regions (F = 48.079) and between male and female plants (F = 6.881; *p* = 0.018) for TPC. The interaction between the sex of plants and the provenance had a significant effect on TFC (F = 6.63; *p* = 0.004) but there was no interaction between the sex of plants and their origins for TPC (F = 1.76; *p* = 0.19).

### 3.2. Antioxidant Activity (AOA)

The results in Figure 3 reveal that, compared to the three other regions, the leaf extract from Bizerte displayed the highest values of TPC and TFC, with 22.04 and 22.78 mg ET/g DM for female and male plant extracts, respectively, followed by Jandouba leaf extracts, with 20.45 mg ET/g DM for female plants and 22.02 mg ET/g DM for male plants, and then Nabeul leaf extracts showed 20.45 and 20.83 mg ET/g DM for female and male plant extracts, respectively. Beja leaf extracts were the lowest, with 19.67 mg ET/g DM for female plants and 19.71 mg ET/g DM for male plants.

These results show that the antioxidant activity was higher in the Bizerte leaf extracts than in other provenances for both sexes, and it was highest in the male plants as opposed to female (*p* < 0.05). The ANOVA test confirmed that the interaction between the sex of plants and the provenance had a significant effect on AOA (F = 6.53; *p* = 0.004) assays. Singh et al. (2016) [21] confirmed that plants which produced a high amount of phenolic compounds possessed high antioxidant activity. These results indicate that the different extracts have significant antioxidant activity which might be responsible for the larvicidal activity studied [22].

### 3.3. Relationship between TPC, TFC and AOA

The correlation between TPC and TFC versus AOA was studied. A positively strong and significant (*p* ≤ 0.01) correlation was observed between TPC, TFC and AOA for the leaf extracts of female plants, with r = 0.854 for TPC and AOA and r = 0.736 for TFC and AOA.

A similar relationship was observed for the leaf extracts of male plants, and our results also established a positive correlation between TPC, TFC and AOA with a Pearson correlation coefficient of r = 0.894 and r = 0.784, respectively.

A positively strong and significant correlation (*p* ≤ 0.01) can be observed between TPC, TFC and AOA. This suggests that phenolic and flavonoid compounds may be responsible for their antioxidant activity.

These results are in agreement with Ghnimi et al. (2014) [9], whose study of the leaf extracts of castor showed a good correlation between phenolic compounds, flavonoid compounds and antioxidant activity.

For the first time, our study on the leaf extracts of *M. annua* revealed a positive correlation between phenolics (TPC and TFC) and antioxidant activity for different provenances, whatever the sex.

### 3.4. LC-MS

The extracts of the male and female *M. annua* leaves contained flavonoids (narcissin, rutin, epicatechin, gallocathechin and epigallocatechin). The comparison to a standard narcissin (isorhamnetin-3-o-rutinoside) of the chromatogram profiles of *Mercurialis* leaf extracts from four Tunisian regions, Bizerte, Beja, Jandouba and Nabeul, under PDA at 350 nm and ESI/MS, proved that narcissin was the main flavonoid in the studied extracts. Our results corroborate with what was advanced by Blanco-Salas et al. (2019) [3], who confirmed the copiousness of the aerial parts of the genus *Mercurialis* in narcissin. The quantification of the different compounds was determined using a calibration curve of each standard (Figure 4 and Table 1).

The plant extracts were rich in narcissin, and those of Bizerte revealed the highest amount, with 885.4 µg/g DM for male plants and 732.78 µg/g DM for female plants, followed by those of Jandouba, with 728.49 µg/g DM (male plants) and 501.38 µg/g DM (female plants), and those of Beja, which displayed the lowest quantity of 320.74 µg/g DM (male plants). The plant extracts were also rich in rutin: 44.18 µg/g DM for Bizerte (male plants), followed by 24 µg/g DM for Jandouba (male plants), then 10.82 µg/g DM for Nabeul (male plants). Beja male plant extracts always had the lowest quantity: 8.11 µg/g DM. The concentration of gallocatechin was important in leaf extractions compared to the other flavonol, with 205.69 and 155.37 µg/g DM in Bizerte male and female plants, respectively. Beja displayed the lowest doses with 57.1 and 42.67 µg/g DM in male and female plants, respectively. This species was also rich in epicatechin and epigallocatechin. Our results were in agreement with those reported in the literature concerning castor leaves belonging to the same *Euphorbeaceae* family. Indeed, Ghnimi et al. (2014) [9] identified epicatechin and rutin in abundance in the aerial parts of some Tunisian provenances of *Ricinus communis* L. Moreover, Aquino et al. (1987) [23], reported that the major phenolic compounds in *M. annua* were narcissin and rutin. These results also revealed the presence in significant quantities of other flavonols such as epicatéchin, gallocatechin and epigallocatechin, which are not yet identified in the recent literature for *M. annua*. The leaf extract analyses showed that the male plants contained more phenolic compounds than the female plants, and that the leaf extracts from Bizerte abounded in phenolic contents compared to other provenances. LC-MS statistical analysis showed that there was a significant difference between regions (F = 13.83 and *p* = 0.002) for the amount of flavonol and between male and female plants (F = 8.149 and *p* = 0.021).

## 4. Larvicidal Activity

The larvicidal effect of the aqueous extracts of the leaves of male and female plants was observed at 6, 12, 24 and 48 h, showing an increasing intoxication of *T*. *confusum* larvae as time (*p* = 0.0001) and concentration (*p* = 0.001) increased (Figure 5). The 0.01 g/mL extract of powder of male plants from the Bizerte region exhibited a high percentage of mortality after 24 h (97%), and 100% mortality was reached after 48 h. This high toxicity was confirmed by the lowest value of LC_50_; LC_50_ = 0.0039 g/mL after 24 h and LC_50_ = 0.0026 g/mL after 48 h. On the other hand, the leaves of the female plants from the Bizerte region showed a higher toxicity than that assessed in the other three regions (Table 2). As detailed in Figure 5, the leaf extract of the Beja provenance displayed the lowest toxicity against larvae, with LC_50_ values for the leaves of male and female plants of 0.00924 g/mL and 0.0157 g/mL, respectively. The leaf extracts of male plants tested on *T*. *confusum* were found more effective as compared to the leaf extracts of female plants. The impact of the sex parameter was highly significant (*p* = 0.001). Our results also underline the highly significant effect of the origin (*p* = 0.002) of the plant material. There was no mortality for the control test.

This study is the first to elucidate the larvicidal activity of the leaves of male and female plants of *M*. *annua* from different regions of Tunisia against *T*. *confusum*. These results corroborate with other research carried out by Al Bachchu et al. (2017) [24] on the larvicidal efficacy of indigenous plant extracts against *Tribolium castaneum* (Herbest) larvae. Aouinty et al. (2006) [17] highlighted the toxic power of aqueous leaf extract of *Ricinus communis* L. (*Euphorbiaceae*) on mosquito larvae. In the same context, several studies [25,26,27] have shown that plant extracts of some species, and at well-defined doses, are able to play an influential role as a bio-larvicide and thus reduce the use of chemicals noxious to the environment and human health.

### Relation between Phenolic Compounds and Larvicidal Activity

Our results reveal for the first time a good negative correlation between the larvicidal activity of aqueous leaf extracts of male and female plants of *M. annua* and the antioxidant activity of methanolic leaf extracts, with a coefficient of correlation of Pearson r = −0.975 for male plants and r = −0.760 for female plants (Figure 6 and Figure 7). As reported by Beatović et al. (2015) [28], the antioxidant capacity of different plant extracts is closely related to their most abundant components. Indeed, narcissin is a natural flavonol glycoside known for its antioxidant activity [29]. Petruk et al. (2017) [30] identified narcissin in immature cladodes of *Opuntia ficus*-*indica* (L.) Mill. and showed that it was responsible for the anti-oxidative stress on human keratinocytes. In addition, in our extracts, gallocatechin occupied a second position after narcissin in terms of abundance, followed by rutin. A previous study demonstrated that rutin significantly blocked the growth and proliferation of larvae of *Aedes aegypti* L. [31]. On the other hand, Sogan et al. (2018) [32] proposed that biological activity could be due to a synergistic effect between the different phenolic compounds.

This study is in agreement with many other studies supporting a negative correlation between the antioxidant activity and biological activity of plant materials [33,34].

## 5. Conclusions

The present phytochemical analysis of leaves of *Mercurialis annua* L. of four Tunisian provenances revealed that, whatever the region, the TPC, the TFC and the antioxidant activity were higher in the extracts of leaves of male plants than in the extracts of leaves of female plants, and also highlighted the abundance of narcissin and rutin in male plants. This result could be used to explain the strong larvicidal activity detected in male leaves against *Tribolium confusum* (Du Val), an insect pest of flour and cereal seeds. This is the first study to focus on the chemical composition of leaf extracts of *Mercurialis annua* L. from Tunisian origins and to evaluate their larvicidal activity on *Tribolium confusum* (Du Val). Indeed, this study clearly showed the importance of aqueous extracts of the leaves of *M*. *annua* as biopesticides against *T*. *confusum*. However, the use of this natural resource must be scientifically approved for sustainable use. In addition, LC-MS analyses revealed the presence of biologically active phenolic compounds responsible for the mortality of *T*. *confusum*. However, more analyses are necessary to purify and identify the molecules responsible for this larvicidal activity and their environmental impact.

## Figures and Tables

**Figure 1 biology-10-00344-f001:**
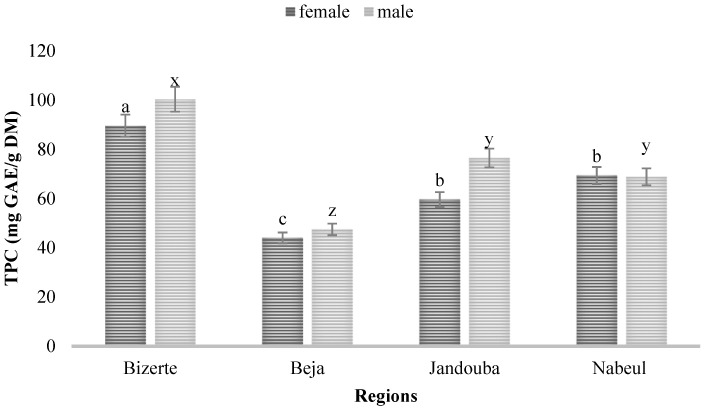
Total phenolic contents (TPC) of leaves of male and female plants of *M*. *annua* from Bizerte, Beja, Jandouba and Nabeul provenances. Results of the Bonferroni test are significantly different at *p* < 0.05. Marked columns not linked by the same letter are significantly different at *p* < 0.05, and each data point is represented by the average of three repetitions ± SD.

**Figure 2 biology-10-00344-f002:**
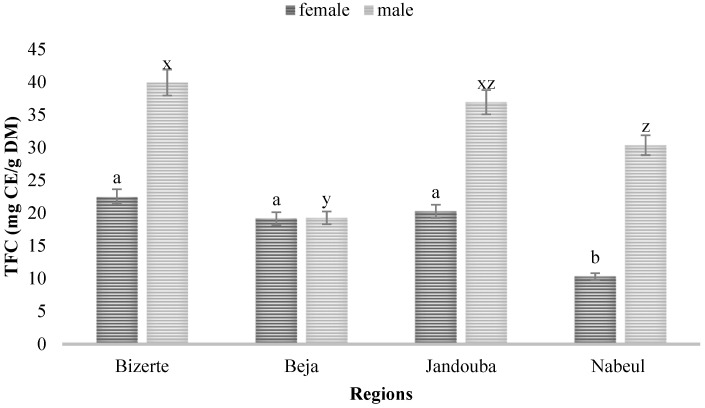
Total flavonoid contents (TFC) of leaves of male and female plants of *M*. *annua* from Bizerte, Beja, Jandouba and Nabeul provenances. Results of the Bonferroni test are significantly different at *p* < 0.05. Marked columns not linked by the same letter are significantly different at *p* < 0.05, and each data point is represented by the average of three repetitions ± SD.

**Figure 3 biology-10-00344-f003:**
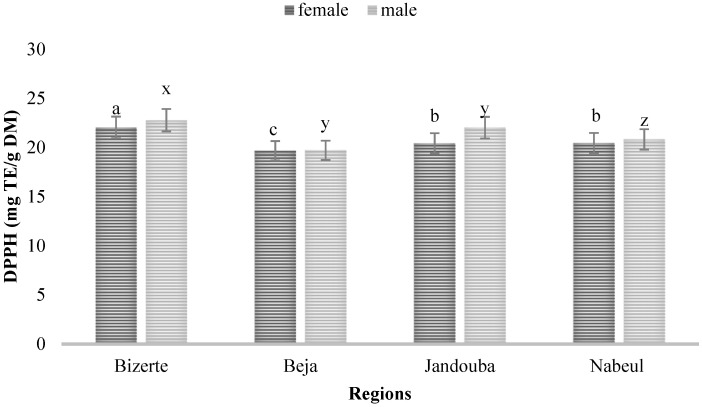
Antioxidant activity (DPPH) of leaves of male and female plants of *M*. *annua* from Bizerte, Beja, Jandouba and Nabeul provenances. Results of the Bonferroni test are significantly different at *p* < 0.05. Marked columns not linked by the same letter are significantly different at *p* < 0.05, and each data point is represented by the average of three repetitions ± SD.

**Figure 4 biology-10-00344-f004:**
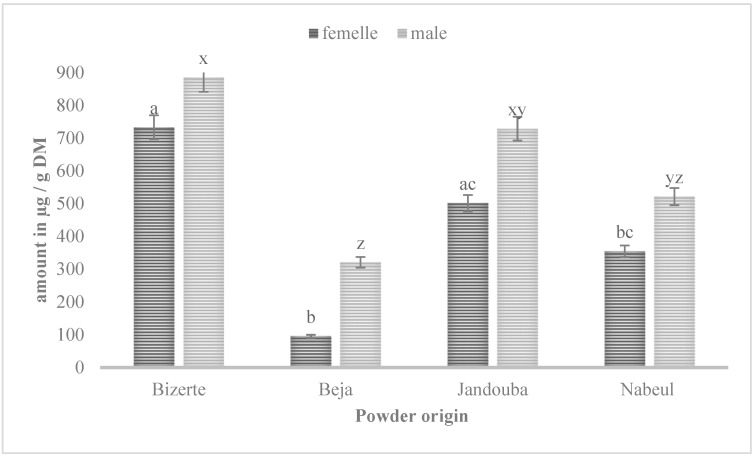
Amount of narcissin in µg/g DM in Bizerte, Jandouba, Beja and Nabeul provenances, for male and female plants. Results of the Bonferroni test are significantly different at *p* < 0.05. Marked columns not linked by the same letter are significantly different at *p* < 0.05, and each data point is represented by the average of three repetitions ± SD.

**Figure 5 biology-10-00344-f005:**
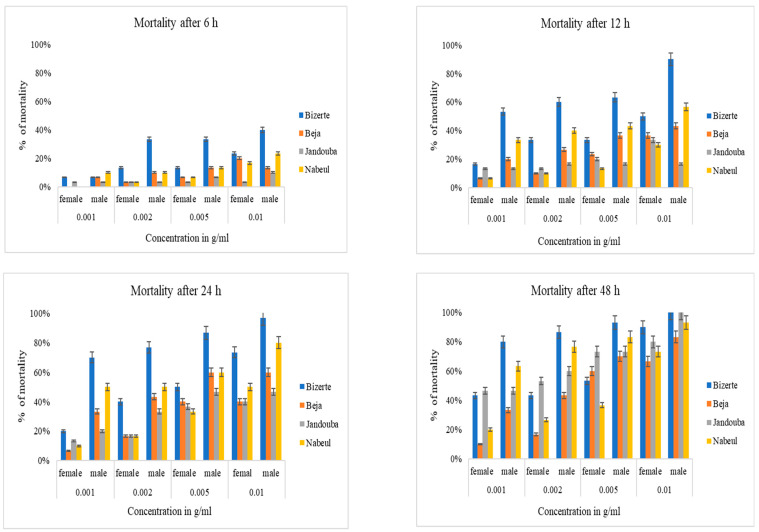
Mortality (%) after 6, 12, 24 and 48 h of *T. confusum* subjected to the different concentrations of the aqueous extract of the leaf powders of the female and male plants of *M. annua* from four provenances.

**Figure 6 biology-10-00344-f006:**
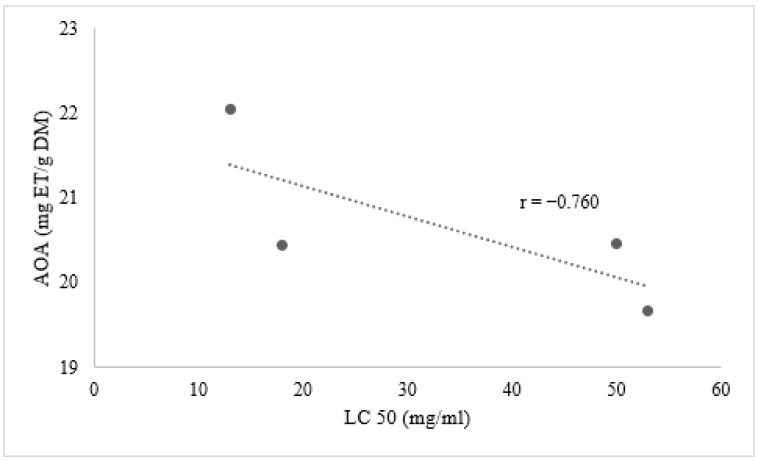
Relationship between antioxidant activity of methanolic leaf extracts and the larvicidal activity of aqueous leaf extracts of female plants of *M*. *annua.*

**Figure 7 biology-10-00344-f007:**
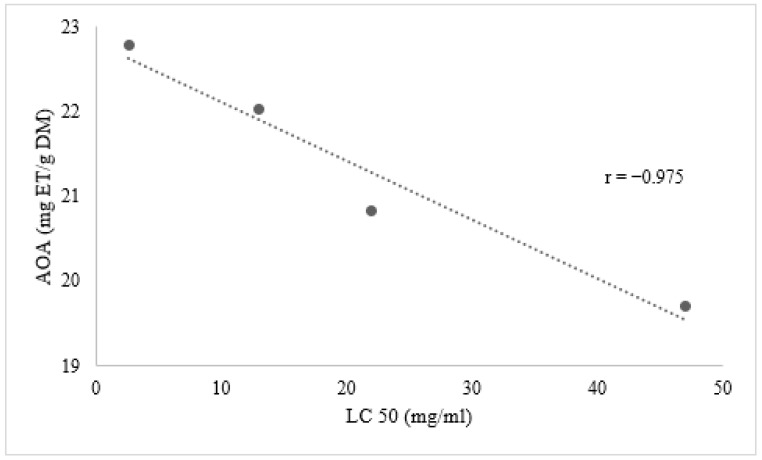
Relationship between antioxidant activity of methanolic leaf extracts and the larvicidal activity of aqueous leaf extracts of male plants of *M*. *annua.*

**Table 1 biology-10-00344-t001:** Concentration (µg/g DM) of flavanol compounds identified in the *M*. *annua* leaves for male and female plants of four Tunisian provenances.

	RT (min)	Amount of Flavonol in µg/g DM							
		Male Plant				Female Plant			
		Bizerte	Beja	Jandouba	Nabeul	Bizerte	Beja	Jandouba	Nabeul
Narcissin	17.5 ± 0.3	885.4 ± 0.001	320.73 ± 0.018	728.48 ± 0.04	521.31 ± 0.038	732.78 ± 0.38	95.02 ± 0.1	501.38 ± 0.73	354.29 ± 0.32
Gallocatechin	7.3 ± 0.2	205.68 ± 0.005	57.1 ± 0.043	181.36 ± 0.004	61.89 ± 0.02	155.37 ± 0.001	42.67 ± 0.005	150.45	59.06 ± 0.02
Rutin	15.4 ± 0.3	44.18 ± 0.016	8.11 ± 0.0002	23.98 ± 0.005	10.82 ± 0.061	18.38 ± 0.004	1.4 ± 0.0003	11.57 ± 0.001	7.4 ± 0.102
Epigallocatechin	9.07 ± 0.2	35.37 ± 0.0007	12.78 ± 0.082	23.68 ± 0.004	20.68 ± 0.006	34.57 ± 0.0008	10.11 ± 0.0012	17.29 ± 0.005	14.63 ± 0.164
Epicatechin	15.3 ± 0.2	27.46 ± 0.04	12.35 ± 0.001	24.57 ± 0.003	18.4 ± 0.0009	21.02 ± 0.003	8.16 ± 0.0006	13.16 ± 0.001	11.01 ± 0.001

Values shown are mean ± SD (*n* = 3).

**Table 2 biology-10-00344-t002:** Larvicidal activity of *M. annua* leaf extract against *T*. *confusum.*

Provenance	Plant Material	Time (h)	LC_50_ g/mL± SE	χ^2^	95% Lower Confidence	95% Upper Confidence
Bizerte	LFP	24	0.0038 ± 0.172	2.969	0.00312	0.00498
	LMP	24	0.00039 ± 0.217	2.237	0.000126	0.00069
	LFP	48	0.0020 ± 0.178	20.439	-	-
	LMP	48	0.00026 ± 0.282	3.406	0.00006	0.0005
Nabeul	LFP	24	0.0022 ± 0.134	0.126	-	-
	LMP	24	0.0014 ± 0.172	5.472	0.00066	0.0022
	LFP	48	0.0049 ± 0.196	10.69	-	-
	LMP	48	0.00046 ± 0.201	1.21	0.00024	0.00099
Jandouba	LFP	24	0.012 ± 0.196	5.045	0.0085	0.0022
	LMP	24	0.0034 ± 0.162	1.711	0.00059	0.024
	LFP	48	0.00135 ± 0.176	0.795	0.00096	0.0042
	LMP	48	0.00132 ± 0.207	16.234	-	-
Béja	LFP	24	0.0157 ± 0.186	1.985	0.0096	0.042
	LMP	24	0.0092 ± 0.173	2.49	0.00013	0.069
	LFP	48	0.0052 ± 0.181	6.859	0.00023	0.034
	LMP	48	0.0022 ± 0.181	0.99	0.0018	0.0028

-: The LC could not be calculated. Heterogeneity factor was used. SE: standard error; LFP: leaves from female plants; LMP: leaves from male plants.

## Data Availability

No new data were created or analyzed in this study. Data sharing is not applicable to this article.

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
