# Peer review of "Phytochemicals, Antioxidant Attributes and Larvicidal Activity of Mercurialis annua L. (Euphorbiaceae) Leaf Extracts against Tribolium confusum (Du Val) Larvae (Coleoptera; Tenebrionidae)"

_biology, 2021, doi:10.3390/biology10040344_

Round 1

Reviewer 1 Report

1 There is no  control  for insecticidal activity test, please add relevant experimental results.

2 The Latin names in the MS are confusing, so please check the MS and abbreviate the rest of the names except for the first one in the text.

3  There are too many spelling and other small errors in the MS that require careful correction.

4  Please adjust the structure of the preface to make it more logical.

Author Response

1 There is no mortality of larvae of T. confusum in control

2 For Latin names and their abbreviations please see the corrections made on the manuscript

3  We corrected the spelling as mentioned in the text (pdf)

4  we have improved the introduction as requested Please look at the manuscript

Reviewer 2 Report

This manuscript has several deficiencies. First of all English could be improved, therefore, the authors may choose to have their paper revised by a competent English speaker before resubmission. Moreover, the way references are presented is confusing and not according to the journal rules (should be in numerical order based on their first appearance in the text). However, what concerns me most about this manuscript is the methodology used to test the toxicity of the extracts that were evaluated. Since the toxicity test used is quite unusual, I would suggest the authors provide a reference for the method used. Also, no information is provided concerning control mortaliry, which is not presented in the graphs. Finally, I feel that the number of larvae used (10) in the tests was low. This, in combination with the low number of replicates (only 3 replicates, rather low I would say), makes the conclusions not so strong.

Author Response

  • To assess the toxicity of our extracts at different concentrations, we were inspired of the method used by Aoiunty et al (2006) and Soha et al (2019), with some modifications.
  • The number of larvae used is 15 and not 10 (typo). because we do not have enough animal material in the lab and there are other students who work with this type of larvae and therefore the number of repetitions will be low too.
  • There is no mortality of larvae of T. confusum in control

Reviewer 3 Report

Statistical analyzes should be run again, verify anova assumptions, include post hoc comparisons, clarify how correlations were made ... Please see comments in pdf version. I found a rather impulsive attempt to find a biological value in the correlations between AOA and larvicidal activities ....

Author Response

For the statistical analyzes, we added the statistical analysis with two control
factors "provenance x sex of plants", to evaluate the interaction between them
and subsequently to validate the Post-hoc test (Bonferoni test used). Please look
at the manuscript

Round 2

Reviewer 1 Report

The work you have done is more detailed and the MS can be aeecpted for publication.

Author Response

(The authors gave the same response as above.)

Reviewer 2 Report

This manuscript has been improved since its last submission. However, I am still highly concerned about the way the toxicity tests were performed. To quote the authors “The toxicity was evaluated by placing 15 T. confusum larvae in glass beaker containing 20 ml of each concentration of each extract”. Moreover, the authors cite previous research studies as references for the toxicity tests that they performed, i.e. Aouinty et al. (2006) and Soha et al. (2019).

The first reference (Aouinty et al. 2006) refers to the evaluation of the larvicidal toxicity of plant extracts against mosquito larvae. As we all know, mosquito larvae are aquatic, therefore this would be the correct way to test the larvicidal activity of the tested compounds, i.e. through the application of the tested substances in the water.

In the second reference (Soha et al. 2019), the authors assessed the larval toxicity of aqueous plant extracts against the freshwater shrimp (Artemia salina), using a similar methodology. To conclude, both references provided, describe toxicity tests against aquatic organisms. Instead, the authors of the present study used a similar experimental procedure (using aqueous solutions) to evaluate the larvicidal toxicity of plant extracts against larvae of a terrestrial species, i.e. the confused flour beetle, Tribolium confusum. To my point of view, toxicity tests were not performed in the correct way.

Moreover, although the number of larvae per replicate was increased since its last submission (15 larvae instead of 10), the number of replicates for a toxicity test still remains low (3 reps). I am still curious, how T. confusum larvae survived up to 48 h in 20 ml of water, as according to the authors there was no control mortality.

I would suggest the authors either exclude the results of the toxicity tests from this work or perform again the toxicity tests in a more appropriate way. There are several references describing how the toxicity of larval extracts could be appropriately performed. A summary of them could be found here: DOI: 10.1007/978-81-322-2006-0_8

Author Response

  • because we do not have enough animal material in the lab and there are other students who work with this type of larvae and therefore the number of repetitions will be low too.
  • sorry we cannot remove the insecticide part, the article does not make sense without this part.